# A Human-Inspired Control Strategy for Improving Seamless Robot-To-Human Handovers

Paramin Neranon * and Tanapong Sutiphotinun

Department of Mechanical and Mechatronics Engineering, Faculty of Engineering, Prince of Songkla University, Songkhla 90112, Thailand; Tanapong_S2539@hotmail.com
* Correspondence: paramin.n@psu.ac.th

**Abstract:** One of the challenging aspects of robotics research is to successfully establish a human-like behavioural control strategy for human–robot handover, since a robotic controller is further complicated by the dynamic nature of the human response. This paper consequently highlights the development of an appropriate set of behaviour-based control for robot-to-human object handover by first understanding an equivalent human–human handover. The optimized hybrid position and impedance control was implemented to ensure good stability, adaptability and comfort of the robot in the object handover tasks. Moreover, a questionnaire technique was employed to gather information from the participants concerning their evaluations of the developed control system. The results demonstrate that the quantitative measurement of performance of the human-inspired control strategy can be considered acceptable for seamless human–robot handovers. This also provided significant satisfaction with the overall control performance in the robotic control system, in which the robot can dexterously pass the object to the receiver in a timely and natural manner without the risk of harm or injury by the robot. Furthermore, the survey responses were in agreement with the parallel test outcomes, demonstrating significant satisfaction with the overall performance of the robot–human interaction, as measured by an average rating of 4.20 on a five-point scale.

**Keywords:** object handover; robot-to-human object handover; human–robot interaction; human–human interaction; hybrid position/force control; impedance control; human-like behavioural control





## 1. Introduction

Recently, robots have been widely developed over the last decades to meet the requirements of improving human–robot interaction (HRI). HRI is a field of study to understand, design and evaluate robotic systems, while interacting with humans [1]. The interaction involves several communication forms and can be mainly categorized into two types: remote interaction (i.e., search and rescue [2–4], military and police [5,6], space exploration [7–9], etc.) and proximate interaction (i.e., assistive robotics and educational robotics [10–14], home or industry [15,16], etc.).

In this research, the field of social robotics and human–robot interaction are highlighted, in which robots are being developed to serve in human assistive capacities. One of the main challenges in close proximity interaction is to efficiently establish a framework for seamless human–robot handovers (HRH). This will allow a human to act jointly with a robot safely and reliably. The researchers [17] claimed that understanding of behavioural control strategies of human–human interaction (HHI) is fundamental to design an effective HRI system. In addition, Neranon et al. [18] also addressed that understanding the principle of human haptic interaction, when two humans work together in a joint effort to complete a shared task, is crucial in designing an effective HRI system.

This section emphasizes the background of handover-based research, since human–human handovers (HHH) are the ultimate benchmark for seamless handovers. A giver is defined as an agent who holds an object and passes it to another. In the physical interaction, a receiver is a particular position that receives the object by starting pulling it. Afterwards,

the receiver takes responsibility for the object throughout the transferring process [19,20]. Much of the relevant HHI research, which has steadily increased to enhance the understanding of HHI behaviour, has been reviewed. Previous studies have examined issues regarding investigating the control characteristics of two humans in a physical handover task, for examples, analysing (1) the influence of velocity [21,22] and trajectory [23,24] on the handover movements, (2) grip forces and load forces in the object exchange [25], (3) grasp points [26,27] and locations [28,29] and finally, (4) configuration [30,31] of the exchanged object. These can be used as conceptual frameworks for designing control systems for cooperative robots to work with other partners by imitating human-based behaviour control strategies.

Sutiphotinun et al. [32] investigated the human behavioural responses in HHH tasks under various conditions. The paradigm findings show human dynamic characteristics, i.e., interactive force analysis, suitable transfer locations between the givers and receivers and the handover sequences. The handover is distinctively categorized into sending-transferring-receiving phases, where the giver agent primarily decides to release the object. This will be useful for developing a robotic behaviour-based approach in seamless HRH in future. Saki Kato et.al [33] analysed how humans (givers and receivers) determine the locations of object transfer during handovers for a good understanding of the mechanisms of human social behaviour. The study also presented a linear model with the influential factors affecting the handover locations. It consists of participant heights, genders and social dominance of the two partners and the distance between both humans.

Chien et al. [34] proposed adaptive coordination strategies for physical HRH. It was based on the identification of HHH strategies regarding how people adapt their handover actions to the workload of their partners. The newly developed approach involving two baselines, namely, (1) proactive and (2) reactive coordination techniques, enabled similarly fluid and effective HRH tasks. Sisbot et al. [35] proposed a robotic navigation planner and a manipulation planning framework by first understanding the human dynamic characteristics of the haptic interaction. This facilitated the robot to autonomously select the best handover location based on context (i.e., accessibility, posture, a field of view and human's safety etc.). This offered safe, legible and socially acceptable paths for HRH tasks. Some researchers [36–38] examined a robotic mechanism guideline to allow the robot to grasp an object based on human-like reaching gestures before handing it over to a human partner. It involved the object's shape and function and the minimization of the risk of an injury occurring during the HRI execution.

Jae-Bong et al. [39] developed a general-purpose software framework for a Human support robot (HSR) service robot. It directly communicated with a human using complex speech commands. This allowed the robot to move smoothly in a confined space and also manipulate various household objects and autonomously pass them to a human. As extensively reviewed, the existing research on the development of HRH has almost underlined the investigation on behavioural strategies and the establishment of a framework for a robotic behaviour-based approach. This allowed the robot to approach a person, reach over an object and hand over it with human-like configurations and motions.

However, there has been little discussion on the crucial challenges of the development of seamless HRI, since a robotic control system is further complicated by the dynamic nature of the human environment. Thus, it necessitates a very careful design and implementation of optimized behavioural control strategies to significantly improve the success rate of effective object handover tasks. This also protects a human partner from the risk of harm or injury by a robot in the cooperative tasks. To solve the limitations as mentioned, therefore, this paper presents the design of a human-inspired control strategy for improving seamless robot–human handovers using the hybrid PID position control and impedance control algorithm. This system will enhance the accuracy of positioning and force regulating abilities in the HRH system and also increase the smoothness of human–robot physical interaction.

This paper details an optimized robotic control implementation for HRH by first understanding haptic dynamic interaction in HHH tasks in terms of interactive force, suitable transfer locations between the givers and receivers and the handover sequences. This knowledge is then used to design a robotic behaviour-based approach, which is the crucial aspect of the development of effective HRH. It is followed by how to develop optimized hybrid position/force control involving PID position control and force impedance control. The robotic control incorporates feedback signals from the HSR-3D-depth sensor, encoders (joint estimators) and multi-axis force/torque sensor attached to the robot wrist joint, respectively. The dynamic mechanical model of the physical HRH was additionally explained. Afterwards, the final sections express the control system evaluations, experimental results of the robotic HRH tasks and finally, conclusion.

## 2. Design of the Robotic Human-Inspired Control Strategy for HRH

### 2.1. Background of Human-Human Handover Strategy

It is crucial to understand the kinematics and dynamics of HHH behaviour to design a conceptual guideline for a robotic human-like controller for robust, behaviour-based, HRH. Consequently, a set of HHH tests was initially conducted as fully detailed in our previous journal article [32]. It can be summarized that the experimental results of HHH intensively addressed (1) investigation of the human handover strategies of the givers and receivers (2) examination of the interactive forces and the transfer locations between the couple and (3) clarification of the mathematical model of the giver's arm, which leads to a better understanding how the giver regulates the bilateral force before releasing the object to be transferred to the receiver.

The tests give the behavioural explanation of HHH in which the HHH process can be interpreted as three distinct phases consisting of (1) sending, (2) transferring and (3) receiving. In the sending session, the giver starts grasping and holding an object and moves to the receiver, who is in the field of the giver's view. Once the handler recognizes a receiver and his/her position, the giver subsequently computes and estimates a suitable object transfer point based on human-relevant parameters. In the meantime, the giver is then moving to the transfer point facing the receiver before naturally handling the object. The handover locations $y$ (height from the ground) is formulated as Equation (1), where $x_1$ and $x_2$ are the giver's and receiver's heights, respectively. Contrastingly, the transfer distance (in the x-axis) along the facing direction was approximately 0.85 m.

$$y = 8.23 + 0.243x_1 - 9.592x_2 - 1.626x_1{}^2 + 1.441x_2{}^2 + 3.189x_1x_2 \tag{1}$$

In the transferring phase, the giver's decision is based on the amount of threshold (interactive) force ($f_{tsh}$), which was significantly proportional to the object mass. The relationship between the $f_{tsh}$ as a function of object mass ($\vartheta_m$) can be exposed in Equation (2).

$$f_{tsh} = -85\vartheta_m{}^3 + 100.75\vartheta_m{}^2 - 35.78\vartheta_m + 6.75 \tag{2}$$

The handover time was approximately 0.46 s. Moreover, to gain more understanding of how the giver regulates the interactive force, the transfer function ($T_G$) of the giver's hand can be expressed as the following equation:

$$T_G = \frac{-2.63 \times 10^{-4}S + 5211}{S^2 - 20.25S + 51.76} \tag{3}$$

These findings disclose the human characteristics of the haptic handover interaction and for the aim of obtaining a conceptual guideline for a robotic human-like control strategy in HRH.

## 2.2. Toyota HSR Platform

An HSR home service as depicts in Figure 1a has been launched by the Toyota Corp. the robot can be controlled intuitively using voice commands or a touchscreen graphical user interface. The core technologies of the HSR comprise speech recognition, object recognition, autonomous navigation and localization, motion planning, task planning, etc. These are adequate to develop HRH in this research. It is realizing that the safety issue is an essential aspect of a robot to collaborate with humans in a safe and natural manner. Obstacle avoidance and a force relaxation mechanism were employed to protect humans from any mistakes or accidents. As the robot has to accommodate a wide variety of household activities, then its cylindrical telescoping body was originally fabricated. It was equipped with a differential pseudo-omnidirectional wheel and a central yaw joint which allow the robot to handle most manipulation tasks with smooth changes in direction [39].

To construct a robot map and map-based self-localization, the robot base has equipped with two types of sensors, i.e., a LIDAR sensor and an inertial measurement unit (IMU). The HSR manipulator arms are made up of a shoulder pitch joint and roll-pitch-roll wrist joints. The robot end-effector has a two-finger parallel gripper (driven by a single motor), which was furnished by a wrist multi-axis force/torque sensor and hand camera. The fingertips were designed to be compliant with the shape of the objects, as it is covered by elastic materials. The robot arm is attached to the body with a three-stage prismatic waist joint, which enables the height of the robot's shoulder to a telescope. The robot's head has been attached by an ASUS Xtion RGBD camera, two RGB cameras, a microphone array and a small display monitor [39–41].

The robot has two onboard computers. The main computer is preserved for Toyota's proprietary code to execute sensor and motor interfaces, motion planning and control and the web-based remote user interface etc. Nevertheless, both computers do not have much computing power for the development of advanced software algorithms. Then, it requires an external computer that can directly communicate with the HSR software platform using Robot Operating System (ROS). The robot software architecture can be categorized into two layers: (1) real-time control and (2) intelligence layers. The first layer executes real-time processing for motor control using motor drivers and servo amplifiers. The second layer deals with analysing or systematically extracting big data such as RGB images and point clouds, in which Linux (Ubuntu) is adopted to run the OS and ROS [39–41]. The HRS software architecture is illustrated in Figure 1b.

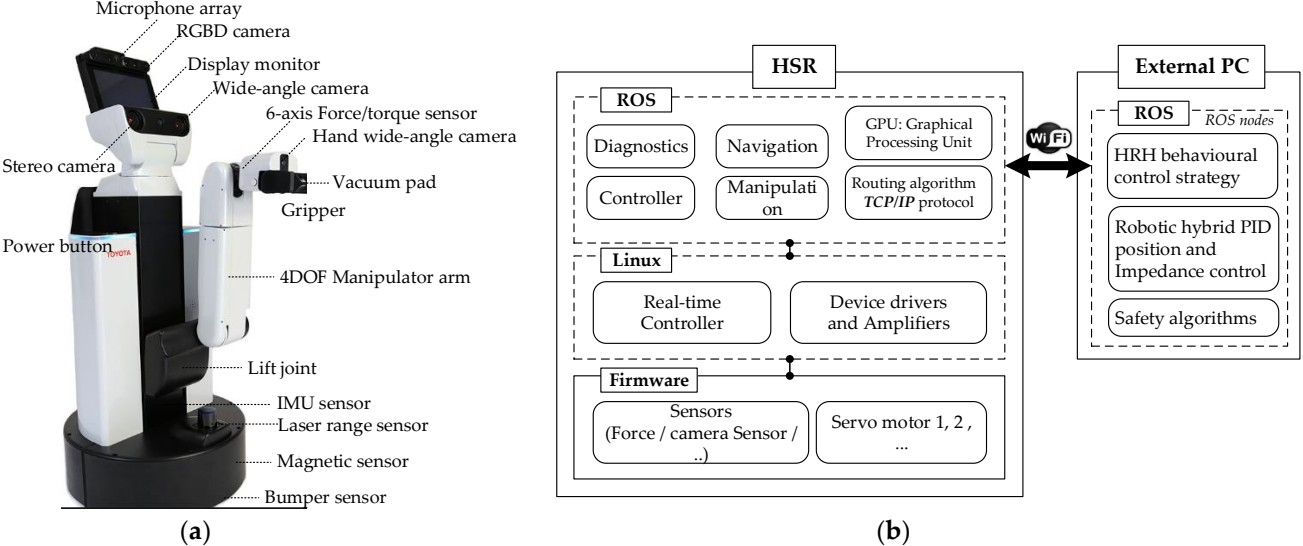

**Figure 1.** (**a**) The hardware equipment of the HRS and (**b**) HRS software architecture (connecting to an external PC).

### 2.3. The Conceptual Framework for Human-Robot Handover

The conceptual outline for HRH presents in Figure 2. Based on the human-like behavioural guideline, it can be summarized that the robot initially takes an order given by a receiver; for example, grasping a bottle of water, etc. Then, the robot moves to the object location and extracts the desired object (i.e., using YOLO). The computational system creates a grasping point of the object based on the centre of gravity. Afterwards, the robot plans the hybrid base/arm trajectory using the hybrid inverse kinematics explained in Section 2.4. Once the robot gripper reaches the object, it starts grabbing it with a proper grasping force measured by force sensors attached to the robot fingertips. The robot moves back to the receiver and simultaneously recognize the receiver and his/her location using the head_pose_estimation package detailed in Section 2.5. Subsequently, the robot moves to a transfer location computed using Equation (1). The transferring phase is activated once the receiver manipulates the hand and grasps the object. In the meantime, the service robot has to regulate the bilateral force before releasing the object to be transferred to the receiver. This computational process incorporates the required maximum interactive force formulated in Equation (2). Then, the final phase involves the receiver taking responsibility for the object and the robot retrieving the arm and returning to the home position.

This paper focuses on the design and development of the robot human-like dynamic control, particularly in the transferring session. This will enable similarly fluid and effective object handover tasks with the receiver. Therefore, for all HRH tests, it can be assumed that the robot correctly holds the required object and stays at a home position already. It is ready to promptly handling the object to a human at a defined transfer location.

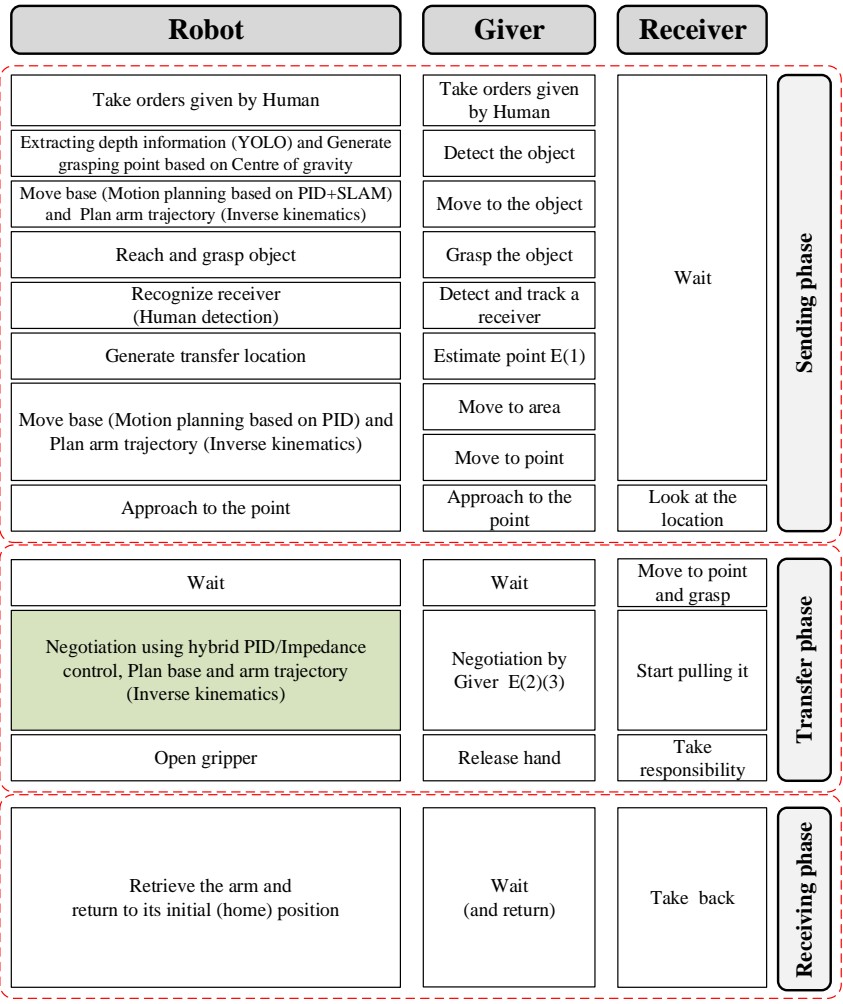

**Figure 2.** Robot handover framework based on the human–human handover.

*2.4. Kinematic Model of the HSR*

This section briefly explains the kinematic model of the HSR, which relates to forward and inverse kinematics. In the case of the forward kinematics, the location and pose of the robot end-effector are controlled by the given links and joints of the HSR robot. The Denavit–Hartenberg parameters evaluation is used to achieve the kinematic chain of the robot rigid bodies. Based on the studies by researchers [39–41], the mechanical schematic diagram of a dual-wheel caster-drive robot base and the robot's joint configuration is shown in Figure 3a. The following notation is used in this analysis: the radius of each wheel ($r$), the angle of the pivot axis ($\theta_H$), the wheel tread ($W$), the displacement between the centre axis and the axle centre ($H$), the inverse kinematics ($f$), the reference value of the tip pose ($T_e$), the degree of freedom of the base ($\theta_2$), the torso lift height ($l_3$), the shoulder pitch angle ($\theta_4$), the length of arm ($l_{52}$), the length of the wrist($l_{82}$) and finally, the joint angles of the arm ($\theta_5, \theta_6, \theta_7$). These are defined in Figure 3b.

The robotic dynamic model is essential for describing the HSR robot's position, orientation and motions of the wheels. It can be used for analysing and synthesizing the overall dynamic behaviour of the robot. The robot itself has three degrees of freedom on the *x-y* plane and can perform forward–reverse movement and left–right movements. As reviewed [39–41], the kinematic equations and relations between robot velocity and various geometry specifications are expressed by the Transformation matrices from linear velocity on the global frame to linear on the robot frame as the following.

$$
\begin{bmatrix} \omega_R \\ \omega_L \\ \omega_H \end{bmatrix} = \begin{bmatrix} \frac{r}{2}cos\theta_H - \frac{rH}{W}sin\theta_H & \frac{r}{2}cos\theta_H + \frac{rH}{W}sin\theta_H & 0 \\ \frac{r}{2}sin\theta_H - \frac{rH}{W}cos\theta_H & \frac{r}{2}sin\theta_H - \frac{rH}{W}cos\theta_H & 0 \\ \frac{r}{W} & \frac{r}{W} & 1 \end{bmatrix}^{-1} \begin{bmatrix} \dot{x} \\ \dot{y} \\ \dot{\theta} \end{bmatrix}. \tag{4}
$$

By assuming that translational and rotational transforms on *x-y-z* axes are represented as $T_t(x, y, z)$, $R_x$, $R_y$, $R_z$ and $R_z$, respectively, forward kinematics of the HSR arm ($T_{HSR}$) can be written as:

$$
T_{HSR} = R_z(\theta_2)T_e(0,0,l_3)R_y(\theta_4)T_e(0,0,l_{52})R_z(\theta_5)R_y(\theta_6)R_z(\theta_7)T_e(0,0,l_{82}). \tag{5}
$$

According to the kinematic of the HSR arm, the base yaw angle ($\theta_2$) can be calculated using the relative wrist position (*x*, *y* and *z*) as:

$$
[x \ y \ z \ 1]^T = fT_e(0,0,-l_{82})[0\ 0\ 0\ 1]^T \tag{6}
$$

$$
\theta_2 = atan2(y, x) \tag{7}
$$

Analysing torso lift height ($l_3$) and shoulder pitch angle ($\theta_4$) gives:

$$
\begin{bmatrix} l_3 \\ \theta_4 \end{bmatrix} = \begin{bmatrix} z - \sqrt{l_{52}^2 - x^2 - y^2} \\ arcsin\left(\sqrt{x^2 + y^2}/l_{52}\right) \end{bmatrix} \text{ or } \begin{bmatrix} l_3 \\ \theta_4 \end{bmatrix} = \begin{bmatrix} z - \sqrt{l_{52}^2 - x^2 - y^2} \\ \pi - arcsin\left(\sqrt{x^2 + y^2}/l_{52}\right) \end{bmatrix}. \tag{8}
$$

Investigating the joint angles of the arm ($\theta_5, \theta_6, \theta_7$) based on the matrix $T_R$ offers:

$$
T_R = T_e(0,0,-l_{52})R_y(-\theta_4)T_e(0,0,-l_3)R_z(-\theta_2)fT_e(0,0,-l_{82}), \tag{9}
$$

$$
\begin{bmatrix} \theta_5 \\ \theta_6 \\ \theta_7 \end{bmatrix} = \begin{bmatrix} atan2(R_{23}, R_{13}) \\ arccos(R_{33}) \\ atan2(R_{32}, -R_{31}) \end{bmatrix} \tag{10}
$$

However, to allow the HSR to interact naturally with a human and to facilitate the dexterous transfer of objects promptly, hybrid inverse kinematics proposed by Yamamoto et al. [41] was implemented. It involves the inverse kinematics for the eight-DOF HSR including the robot base with three DOF and the arm with five DOF. The hybrid inverse

kinematics ($f$) for the HSR can be formulated using the robot tip pose ($T_e$) as in the following equation:

$$(\theta_0, \theta_1, \theta_3, \theta_5, \theta_6, \theta_7) = f(T_e, \theta_2, \theta_4) \tag{11}$$

Given $\theta = (\theta_0, \theta_1, \theta_2, \theta_3, \theta_5, \theta_6, \theta_7)$ and the desired reference joint angles ($\theta^{ref}$) of $\theta$ and then the evaluation function ($V$) can be identified as:

$$V\left(\theta_2, \theta_4, T_e, W, \theta^{ref}\right) = \left|\left| W\left(\theta^{ref} - \theta\right) \right|\right| \tag{12}$$

The evaluation function ($V$) can be minimized using $\theta_2$ and $\theta_4$ and expressed as:

$$\arg V_{min}\left(\theta_2, \theta_4, T_e, W, \theta^{ref}\right). \atop \theta_2, \theta_4 \tag{13}$$

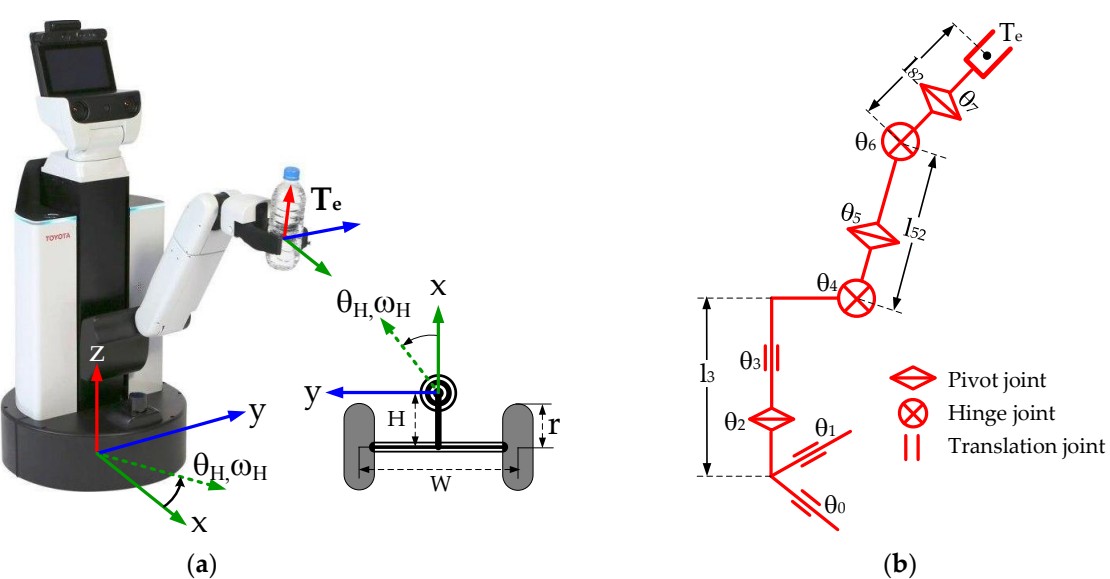

**Figure 3.** (**a**) The dual-wheel caster-drive robot base and (**b**) the robot join configuration.

### 2.5. Coordinate Registration in the HRH System

This section proposes all coordinate registration for the HRH system. It consists of the robot base frame (robot reference frame) [R], the robot end-effector frame [A], object (held by the robot) frame [T], RGBD camera frame [C], human frame [H] and handover location frame [L]. All relative frames in the HRH system are drawn in Figure 4. It is to be noted that all the reference frames are in Cartesian coordinates. According to the human frame [H] can be located by a tracking control based on the head_pose_estimation package [42] as mentioned previously. A human head is identified based on a personalized template by the spherical kernel using a supervised learning algorithm, namely a discriminative random regression forests (DRRF) technique. The human nose tip vector is a crucial aspect that can describe the changes of the head pose neatly. Subsequently, the nose tip frame is appropriately used as the human reference frame [H]. Then, it can be noted that the handover location frame [L] is associated with the human (nose tip) frame [H].

The frame relationship between the object held by the robot and the human hand is the chief requirement implemented to successfully control the robot movements in the HRH tasks. The Transformation matrix has been investigated to identify the posture of the object [T] held by the HSR end-effector [A] relative to the robot reference frame [R]. It can be implied that the robot should manipulate the held object to the receiver naturally at the handover location (incorporating Equation (1)). The Transformation matrix of the handover location frame [L] relative to the robot base frame [R] can be determined based on (1) the

[L] referenced by the [C] and (2) the [L] referenced via [R]. Hence, their relationship can be written as Equation (14).

$$
{}_T^R T = {}_A^R T \cdot {}_T^A T = {}_C^R T \cdot {}_H^C T \cdot {}_L^H T \cdot {}_T^L T \tag{14}
$$

where ${}_j^i T$ denotes the coordinate transformation from the *j* frame relative to the *i* frame. The ${}_T^R T$ can be solved using the robot forward kinematics equations. The ${}_H^C T$ can be measured by the tracking system. The ${}_C^R T$ can be adopted from the robot configuration and ${}_L^H T$ can be calculated using Equation (1) conducted by the HHH experiments. Finally, the ${}_T^L T$ can be computed using the following equation:

$$
{}_T^L T = \left( {}_L^H T \right)^{-1} \cdot \left( {}_H^C T \right)^{-1} \cdot \left( {}_C^R T \right)^{-1} \cdot {}_A^R T \cdot {}_T^A T \tag{15}
$$

The transformation matrix of the [L] frame relative to the [R] can be expressed by Equation (16).

$$
{}_L^R T = {}_A^R T \cdot {}_T^A T \cdot \left( {}_T^L T \right)^{-1} \tag{16}
$$

The relationship is subsequently implemented on the real-time hybrid PID position and impedance control of the robot compliant motions in HRH.

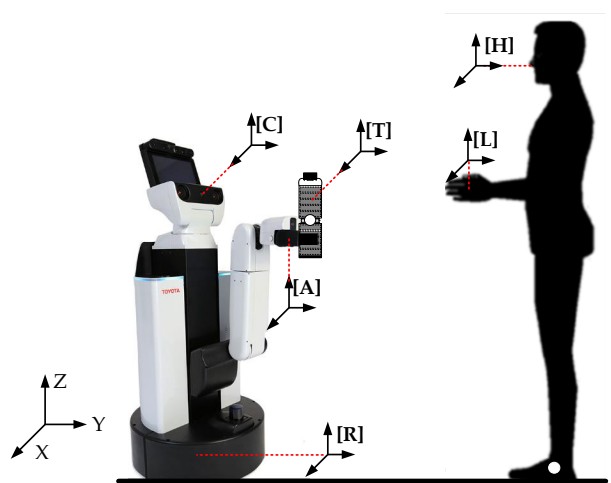

**Figure 4.** Reference frames of the HRH system.

*2.6. Robotic Hybrid Position/Force Control Strategy for the HRH Tasks*

Robot force control is a fundamental requirement in the achievement of the control of the robot's real-time path in any physical HRI task. It has been developed in the past three decades, using for example force, torque and visual feedback to operate robots to suitably participate in unstructured environments. Raibert and Craig [43] introduced a new method for robot force control based on two control loops, consisting of combined position and force control systems, namely hybrid position/force control. One of the crucial advantages of using the hybrid control is that the position and force information is analysed synchronously and independently. These two outputs are combined in the final stage before being converted to drive the robot actuators using joint torque control [44].

In this research, hybrid PID position/impedance control was implemented. This is due to the fact that PID position control is the most widely used and reliable method for controlling a robot manipulator position. Additionally, a method for the automatic tuning of PID controllers in a closed loop can be estimated using a transfer function model. However, the PID controller may not suitable for force control because the derivative term ($K_d$) is too sensitive to noise. This could induce a destabilizing impact on the robot force control system [45]. Interestingly, Rahman et al. [46] proposed the mathematical models of the human hand while interacting with a HHH task. The human applied force is formulated by the human arm impedance which is made up of mass, stiffness and damping factor,

respectively. Consequently, to imitate the real-world human force dynamics, impedance control, which efficiently controls both the motion of the robot and its contact forces, was selected in the robotic force control loop.

Figure 5 illustrates the hybrid PID position/impedance control scheme, which was designed based on the study by Vukobratovic et al. [47]. It is made up of two control loops each of which has an individual information sensor system, i.e., (1) multi-axis force/torque sensor used to detect the force applied to the HSR robot end effector and (2) the rotary encoders used to measure the robot joint positions. $X_R$ and $F_R$ represent the references of the position and force control schemes, which can be generated using handover experimental results, as explained in Equations (1) and (2). The outcomes from the position ($X_P$) and force control ($X_F$) loops are fused together as a set of incremental displacements ($X_u$), namely, the hybrid position/force control, as exhibited in Equation (17). The parameters $X_E$ and $F_E$ are the errors of the position in Cartesian coordinate and force control algorithms, respectively. The matrix $S$ represents the compliance selection matrix used to specify the number of DOF in the robotic control system, where $A$ and $B$ represent the robotic position and force controllers, respectively.

$$X_u = X_P + X_F = A[(S \times X_E)] + B[((1 - S) \times F_e)] \tag{17}$$

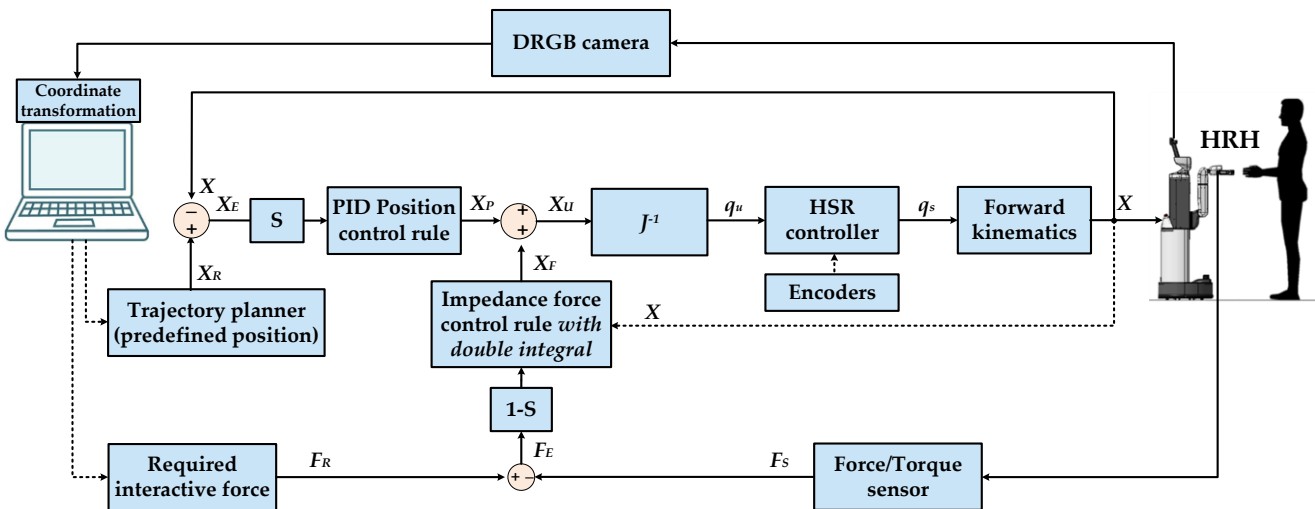

**Figure 5.** Hybrid PID position/impedance force control designed for the HSR in the HRH tasks.

The positions of the robot in joint space ($q_u$) can be computed based on an inverse Jacobian matrix as expressed in the following equation.

$$q_u = J(q)^{-1} X_u \tag{18}$$

The position feedback of the robot in Cartesian space ($X$) is adopted the forward kinematic function based on the position feedback of the robot joint space detected by the encoders ($q_s$) as formulated in Equation (19).

$$X = f(q_s) \tag{19}$$

According to the robot control design, a PID controller was implemented on the robot position control. An incremental discrete-time PID control algorithm with a sampling period ($\tau$) and the discrete-time interval $k$ was implemented. The robotic position control scheme is written as:

$$X_P(k) = X_P(k-1) + (S)\left[k_p(X_E(k) - X_E(k-1)) + k_i(X_E(k)) + k_d(X_E(k) - 2X_E(k-1) + X_E(k-2))\right] \tag{20}$$

Additionally, the robot has to interact with a human in HRH, in which the muscles of the human arm can be mechanically represented in a mass-spring and damper system [46]. To facilitate the robotic impedance behaviour, the impedance control approach is therefore executed, which is not only used to track a motion trajectory alone but rather to regulate the mechanical impedance ($Z_m$) of the robot. The impedance control can be exposed as:

$$F_E = M_d\left(\ddot{X}_F - \ddot{X}\right) + B_d\left(\dot{X}_F - \dot{X}\right) + K_d(X_F - X) \tag{21}$$

The real-time trajectory of the robot end-effector can be controlled by acceleration as the following equation; however, the acceleration output incorporates a double integral function before being transmitted to the robot.

$$\ddot{X}_F = \ddot{X} + M_d^{-1}\left[F_E - B_d\left(\dot{X} - \dot{X}_F\right) - K_d(X - X_F)\right] \tag{22}$$

where $M_d$, $B_d$ and $K_d$ represent a designed inertia matrix, damping matrix and stiffness matrix, respectively, which incorporate with $F_E$.

### 2.7. Dynamic Mechanical Model of the Physical Human-Robot Handover

This section summarized a mathematical dynamic model of the HRH system. Presumably, the overall interaction can be represented as sub-equivalent-lumped-mass systems as depicted in Figure 6. The HSR can be modelled by the rigid body model ($m_R$) associated with effective viscous damping ($b_R$) to ground and driven by force $F_R$ created by the hybrid control algorithm. The robot hand (gripper), which holds the rigid object mass $m_O$, has the impedance parameters as $m_h$, $k_h$ and $b_h$, respectively. The force sensor attached to the hand wrist can be mechanically presented as a spring ($k_s$) and dashpot ($b_s$) system. The impedance parameter notation of the human hand comprises the hand mass ($m_H^*$), the viscous damping ($b_H$) and stiffness ($k_H$), respectively. In addition, the displacements of the masses ($m_R$, $m_h$ and $m_H$) are, respectively, defined as $x_R$, $x_h$ and $x_H$, in which it can be assumed as: $m_H = m_O + m_H^*$. The state-space equations based modelling representing the overall HRH system can be formulated as:

$$\frac{d}{dt}\begin{bmatrix} x_R \\ \dot{x}_R \\ x_h \\ \dot{x}_h \\ x_H \\ \dot{x}_H \end{bmatrix} = \begin{bmatrix} 0 & 1 & 0 & 0 & 0 & 0 \\ -\frac{k_s}{m_R} & -\frac{(b_R+b_s)}{m_R} & \frac{k_s}{m_R} & \frac{b_s}{m_R} & 0 & 0 \\ 0 & 0 & 0 & 1 & 0 & 0 \\ \frac{k_s}{m_h} & \frac{b_s}{m_h} & -\frac{(k_h+k_s)}{m_h} & -\frac{(b_h+b_s)}{m_h} & \frac{k_h}{m_h} & \frac{b_h}{m_h} \\ 0 & 0 & 0 & 0 & 0 & 1 \\ 0 & 0 & \frac{k_h}{m_H} & \frac{b_h}{m_H} & -\frac{(k_h+k_H)}{m_H} & -\frac{(b_h+b_H)}{m_H} \end{bmatrix}\begin{bmatrix} x_R \\ \dot{x}_R \\ x_h \\ \dot{x}_h \\ x_H \\ \dot{x}_H \end{bmatrix}$$

$$+ \begin{bmatrix} 0 & 0 & 0 & 0 & 0 & 0 \\ \frac{1}{m_R} & 0 & 0 & 0 & 0 & 0 \\ 0 & 0 & 0 & 0 & 0 & 0 \\ 0 & 0 & 0 & 0 & 0 & 0 \\ 0 & 0 & 0 & 0 & 0 & 0 \\ 0 & 0 & 0 & 0 & 0 & 0 \end{bmatrix}\begin{bmatrix} F \\ 0 \\ 0 \\ 0 \\ 0 \\ 0 \end{bmatrix} \text{ and } y = \begin{bmatrix} 1 & 0 & 0 & 0 & 0 & 0 \\ 0 & 0 & 1 & 0 & 0 & 0 \\ 0 & 0 & 0 & 0 & 1 & 0 \end{bmatrix}\begin{bmatrix} x_R \\ \dot{x}_R \\ x_h \\ \dot{x}_h \\ x_H \\ \dot{x}_H \end{bmatrix} + [0]F \tag{23}$$

After taking the Laplace transforms of the state space equations above, the transfer matrices are then given by:

$$G_1(s) = \frac{x_R(S)}{F_R(S)} = \frac{A}{D}, \; G_2(s) = \frac{x_h(S)}{F_R(S)} = \frac{B}{D}, \; G_3(s) = \frac{x_H(S)}{F_R(S)} = \frac{C}{D} \tag{24}$$

where

$$
\begin{aligned}
A = \quad & (k_h k_s + k_h k_H + k_s k_H + b_h k_s S + b_s k_h S + b_h k_H S + b_H k_h S + b_s k_H S + b_H k_s S + \\
& b_h b_s S^2 + b_h b_H S^2 + b_s b_H S^2 + b_h m_h S^3 + b_h m_h S^3 + b_h m_H S^3 + b_H m_h S^3 + b_s m_H S^3 + \\
& k_h m_h S^2 + k_h m_H S^2 + k_H m_h S^2 + k_s m_H S^2 + m_h m_H S^4) \\
B = \quad & ((k_s + S b_s)(k_h + k_H + b_h S + b_H S + m_H S^2)) \\
C = \quad & ((k_h + S b_h)(k_s + S b_s)) \\
D = \quad & (k_h k_s k_H + b_h b_R b_s S^3 + b_h b_R b_H S^3 + b_h b_s b_H S^3 + b_R b_s b_H S^3 + b_h b_R k_s S^2 + \\
& b_R b_s k_h S^2 + b_h b_R k_H S^2 + b_R b_H k_h S^2 + b_h b_s k_H S^2 + b_h b_H k_s S^2 + b_s b_H k_h S^2 + \\
& b_R b_s k_H S^2 + b_R b_H k_s S^2 + b_h b_R m_h S^4 + b_h b_s m_h S^4 + b_h b_s m_R S^4 + b_h b_R m_H S^4 + \\
& b_h b_H m_R S^4 + b_R b_H m_h S^4 + b_h b_s m_{Hw} S^4 + b_s b_H m_h S^4 + b_R b_s m_H S^4 + b_s b_H m_R S^4 + \\
& b_R k_h m_h S^3 + b_h k_s m_h S^3 + b_s k_h m_h S^3 + b_h k_s m_R S^3 + b_s k_h m_R S^3 + b_h k_H m_R S^3 + \\
& b_R k_h m_H S^3 + b_R k_H m_H S^3 + b_w k_g m_r S^3 + b_g k_s m_w S^3 + b_s k_g m_w S^3 + b_s k_w m_g S^3 + \\
& b_w k_s m_g S^3 + b_r k_s m_w S^3 + b_s k_H m_R S^3 + b_H k_s m_R S^3 + b_h m_h m_R S^5 + b_h m_R m_H S^5 + \\
& b_R m_h m_H S^5 + b_H m_h m_R S^5 + b_s m_h m_H S^5 + b_s m_R m_H S^5 + k_h k_s m_h S^2 + k_h k_s m_R S^2 + \\
& k_h k_H m_R S^2 + k_h k_s m_H S^2 + k_s k_H m_h S^2 + k_s k_H m_R S^2 + k_h m_h m_R S^4 + k_h m_R m_H S^4 + \\
& k_H m_h m_R S^4 + k_s m_h m_H S^4 + k_s m_R m_H S^4 + m_h m_R m_H S^6 + b_R k_h k_s S + b_R k_h k_H S + \\
& b_h k_s k_H S + b_s k_h k_H S + b_H k_h k_s S + b_R k_s k_H S
\end{aligned}
$$

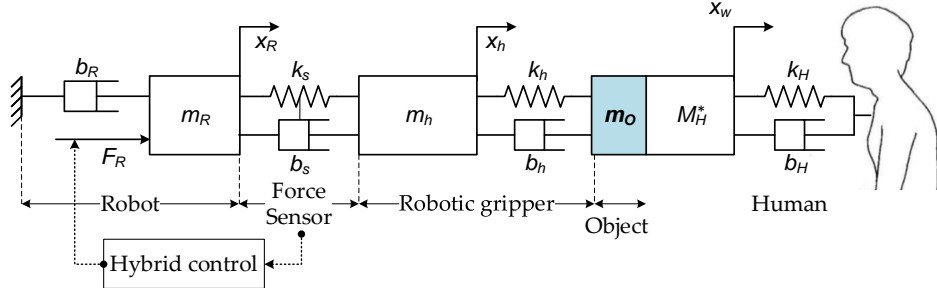

**Figure 6.** Mechanical schematic of the HRH based on a second-order-lumped-mass model.

### 2.8. Safety in HHH

One of the principal challenges in the development of the HRH task is a safety issue since any failure might become very critical for a human co-worker. Several studies have investigated relative approaches to safety and reliability issues in HRI, i.e., human factors, risk assessment, hazard analysis and technologies for HRI safety, etc. Matthias et al. [48] presented an assessment of the associated risks for possible incidental contact events between a collaborative robot and a human worker based on the guidance published in ISO/TS 15066. The ISO/TS 15066 provides standards of safety and operation for the complex protection schemes required for robots. Many researchers [49–51] proposed conceptual designs to improve safety issues in HRI, such as applying appropriate emergency stops which can be activated as soon as any error occurs, or implementing a validation phase based on control software to facilitate an effective real-time control system etc.

In the robotic system, a safeguarded zone was introduced to conduct a safety strategy that can prevent unauthorised humans from entering the test area. The speed of the HSR movement allowed in the HRI tests was limited and the robot working area was also optimized to minimize the risk of an injury. A timeout, which indicates the period of time allowed for executing a specified task, was assigned for the serial and TCP/IP communication. Safety upper-limit force ($f_{lim}$) was defined. Based on the HHH study [32], the $f_{lim}$ was recommended to initially set at 10 N indicating the amount of magnitude of maximum force is taken into account when the *handler* decides to release the object to be transferred naturally. If the exerted (interactive) force is greater than the limited force, the robot will be immediately terminated. Moreover, the stand-alone emergency stop buttons can be manually activated by a human operator when accidents are detected.

## 3. Results and Discussion

The HHH cooperation has provided a better understanding of what is required in the development of an appropriate set of human-like behaviour for the robot control strategy.

This can facilitate the dextrous HRH tasks in a safe and effective way. In this research, hybrid PID position and impedance control has been developed and implemented to improve the stability of the robot position and force control. Nevertheless, the key challenge is to suitably design the optimized PID and impedance control algorithms of the HSR, which interacts with the complicated dynamic nature of the human in the object handovers. Subsequently, this section details the evaluation of the performance of the robot control system to facilitate the robot to transfer the object to the human receiver in a safe, timely and natural manner.

### 3.1. Optimization of the Robotic PID Position Control

To successfully adopt an appropriate PID controller, the control gains ($k_p$, $k_i$ and $k_d$) can be estimated using the Ziegler–Nichols tuning method incorporated with the robot transfer function of the proposed dynamic model in Section 2.7. However, it is complicated to estimate each model parameter in the dynamic equations and then the MATLAB system identification was consequently applied. One of the effective parametric models identifying system transfer functions which are recommended by Ruslan et al. [52] is the Autoregressive Moving Average eXogenous (ARMAX). The ARMAX is one of the most robust techniques and is normally used to study complex dynamical systems in time-series analysis.

The Z–transform is reasonable for analysis of practical model identification, which presents the relationship between the eigenvalues and the poles of the ARMAX-based estimated model. The model [53] can be formulated as follows:

$$y(t) + a_1 y(t-1) + \cdots + a_{n_a} y(t - n_a) = b_1 u(t-1) + \cdots + b_{n_b} u(t - n_b) + e(t) \\ + c_1 e(t-1) + \cdots + c_{n_c} e(t - n_c) \tag{25}$$

where the observed input and output are represented as $u(k)$ and $y(k)$. The noise signal moderated into the system is $e(k)$.

The Matlab Identification Toolbox was implemented to appropriately estimate the model unknown ARMAX parameters by selecting a set of a polynomial of $n_a$, $n_b$ and $n_c$ to minimize the forecasting errors and to obtain an effective model. A set of preliminary experiments involving a robot position control system has been conducted. The time-series information required in the system identification is made up of (1) robot position target, namely the ARMAX input and (2) the measured robot position, namely the ARMAX output. Presumably, the range of the robot forward translation movement is approximately 1 meter. Additionally, to present the system-parameter estimation technique under consideration adequately, the model validation finally verifies a proposed identified model. The discrete-time ARMAX model was identified and its results are given in Equation (26). The set of minimized polynomial functions is made up of $n_a : 2$, $n_b : 2$ and $n_c : 2$, respectively.

$$A(z) = 1 + a_1 z^{-1} + \cdots + a_{n_a} z^{-n_a} = 1 - 1.993\, z^{-1} - 0.9934 z^{-2} \\ B(z) = b_0 + b_1 z^{-1} + \cdots + b_{n_b} z^{-n_b} = 0.0001068 z^{-1} - 9.555\text{e} - 05 z^{-2} \\ C(z) = 1 + c_1 z^{-1} + \ldots + a_{n_a} z^{-n_c} = 1 - 0.9241 z^{-1} - 0.02802 z^{-2} \tag{26}$$

It is to be noted that the MATLAB discrete-time to continuous-time model conversion was then applied to determine the robotic transfer function into a continuous-time transfer function ($HSR_{TF}$) as formulated in Equation (27).

$$HSR_{TF} = \frac{1.036\text{e} - 05 s^2 + 0.002733 s + 0.03113}{s^2 + 1.666 s + 0.7003} \tag{27}$$

Model validation was carried out in the final stage of the model identification process to verify the simulated model. Several model validation techniques were then employed and the results express as follows: (1) percentage of best fit ($R^2$) was approximately 100%, (2) mean squared error (MSE) was $1.069 \times 10^{-10}$ and finally, (3) final prediction error (FPE) was $1.077 \times 10^{-10}$, respectively. Using the predicted model indicates that the actual and

estimated data have high similarity in terms of the system responses. Then, it is considered acceptable for the HSR representation model.

After executing Ziegler Nicholas PID tuning in Matlab by providing the numerator and denominator coefficients of the HSR mathematical model in the Laplace domain, it gives gain parameters the critical gain $k_c$ and ultimate period $\tau_c$. Consequently, it can compute the gain values of $k_p$, $k_i$ and $k_d$. Nevertheless, the tuning technique still needs an operator's experience for fine-tuning until the system specification is achieved. This offers optimized PID gains as $k_p$, $k_i$ and $k_d$ of 0.24, 0.13 and 0.02, respectively. To evaluate the closed-loop stability of the robot system, Figure 7 illustrates the test results and frequency response analysis, whereas the robot moved forwards to the 1 m target. It can be summarized that a set of the optimized PID gains demonstrates significant satisfaction with the overall control performance in the robotic control system and this is subsequently used in the robot hybrid control approach.

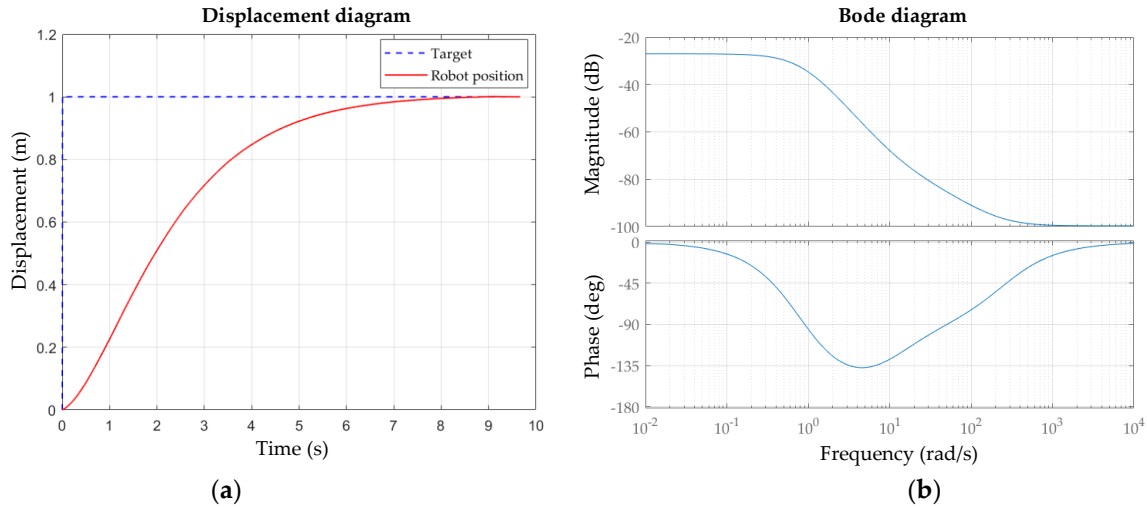

**Figure 7.** (**a**) Robot displacement profile in the HRH test and (**b**) Bode magnitude and phase plot.

### 3.2. Evaluation of the Robotic Impedance Control Based on the HRH

This section proposes how to achieve the appropriate impedance force control based on the performance evaluations of the robotic dynamic response in the human-to-robot handovers. According to the test procedure, a set of 20 participants has undertaken three repetition sets of each handover test under the required variable conditions. They are defined as a receiver agent and have to grasp the object using one hand and twisting the object is not allowed. The object was fabricated as a bottle-shaped object with 60 mm diameter, 270 mm length and a total mass of 1.25 N. The HRH experimental apparatus is shown in Figure 8. The HSR is required to grasp an object and move facing to the object handover location, while the human performs standstill in comfortable positions on opposite sides of the robot. Once the object is manipulated to the transfer location the human is then allowed to grab the object and start naturally pulling the object before taking the responsibility for the object finally. In the meantime, interactive force ($F_{int}$) profiles against time ($t$) presenting how the robot giver regulates the bilateral force before releasing the object to be transferred to the receiver was measured in real-time.

The circumstances affecting the robot dynamic behaviour compose of spring stiffness ($K_d$) and damping factor ($B_d$), whereas $M_d$ was strictly specified as a constant parameter of 0.25 Kg. Initially, a set of preliminary tests was undertaken to establish the relationship between the interactive force and handover time under various spring stiffness ($K_d$) and damping factor ($B_d$) values. The goals of the experiments are to understand how the influential factors affect the interactive forces in the HRH system and to determine the ranges of $B_d$ and $K_d$ used in the substantive HRH. The test outcomes were demonstrated in Table 1. According to human active comfort, the impedance control system should

have lower stiffness and damping coefficient values. Conflictingly, in terms of active safety, it should have lower stiffness but a higher damping coefficient. Based on the recommendations of the preliminary study, as the humans generally felt comfortable in the natural physical HRH, $B_d$ should be set as 50 and 75 (NmS/rad) and the range of $K_d$ should be varied from 0 to 400 (Nm/rad), respectively.

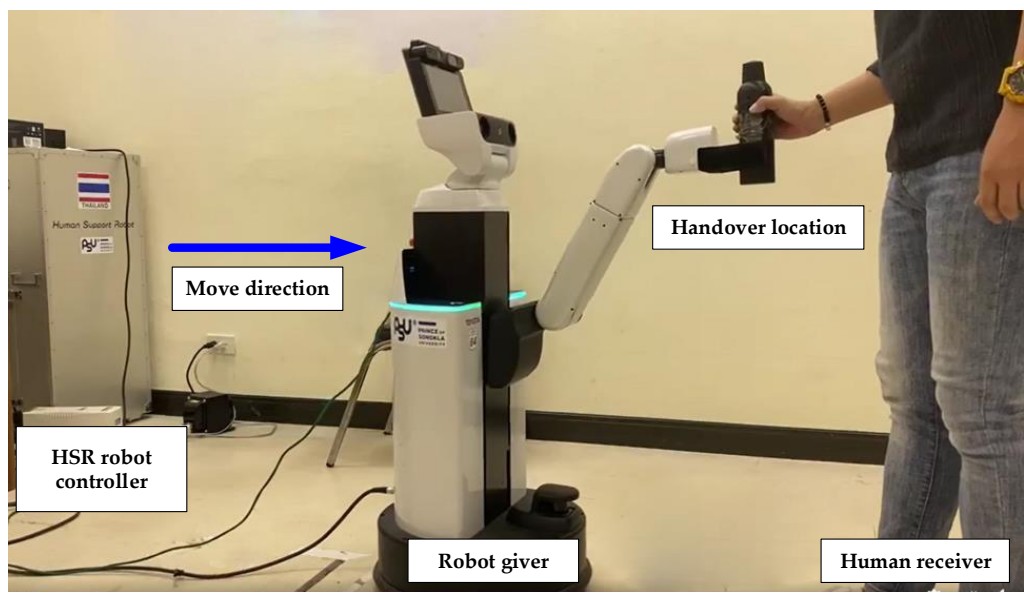

**Figure 8.** Experimental setup of the robot-to-human object handover system.

**Table 1.** Comparisons of the average interactive force ($F_{int}$) and time ($t$) parameters, while individually varying the damping factor ($B_d$) and spring stiffness ($K_d$), respectively.

| Varied Damping Factor ($B_d$) | | | | Varied Spring Stiffness ($K_d$) | | | |
|---|---|---|---|---|---|---|---|
| $B_d$ (NmS/rad) | $K_d$ (Nm/rad) | $F_{int}$ (N) | $t$ (s) | $B_d$ (NmS/rad) | $K_d$ (Nm/rad) | $F_{int}$ (N) | $t$ (s) |
| 25 | 0 (N/A) | 1.19 | 2.14 | 0 (N/A) | 50 | 0.97 | 3.03 |
| 50 | 0 (N/A) | 1.16 | 2.22 | 0 (N/A) | 100 | 1.00 | 2.99 |
| 75 | 0 (N/A) | 1.12 | 2.32 | 0 (N/A) | 150 | 1.04 | 2.92 |
| 100 | 0 (N/A) | 1.09 | 2.41 | 0 (N/A) | 200 | 1.07 | 2.90 |
| 125 | 0 (N/A) | 1.05 | 2.49 | 0 (N/A) | 300 | 1.16 | 2.84 |
| 150 | 0 (N/A) | 1.00 | 2.61 | 0 (N/A) | 400 | 1.20 | 2.80 |

The substantive HRH experiments were successfully carried out by the 20 selected subjects. A comparison of the average interactive force and handover time under the several conditions based on the HRH tasks is given in Table 2. After careful observation with regard to the HRH test results, it was complicated to select the set of optimized impedance parameters as its nature necessitates human judgments. After asking the participants, two desirable choices of the impedance parameters were selected based on the recommendations, which consist of Set 1: $B_d$(50), $K_d$ (200) and $M_d$(0.25) and Set 2: $B_d$(75), $K_d$(50) and $M_d$(0.25), as highlighted in Table 2. A questionnaire technique was subsequently employed to gather information from the human participants. Rating scale questions were introduced and specified with the five-point rating scale from 1 (very dissatisfied) to 5 (very satisfied). The two questions were used to ask the participants after participating in the HHH and HRH based on both impedance parameter sets as follows:

(1) How do you compare the qualitative performance of the HHH to that of the HRH using the parameter Set 1?

(2)  How do you compare the qualitative performance of the HHH to that of the HRH using the parameter Set 2?

**Table 2.** Comparisons of the average interactive force ($F_{int}$) and time ($t$) parameters, while varying both damping factor ($B_d$) and spring stiffness ($K_d$).

| Varied Damping Factor ($B_d$) | | | | Varied Spring Stiffness ($K_d$) | | | |
|---|---|---|---|---|---|---|---|
| $B_d$ (NmS/rad) | $K_d$ (Nm/rad) | $F_{int}$ (N) | $t$ (s) | $B_d$ (NmS/rad) | $K_d$ (Nm/rad) | $F_{int}$ (N) | $t$ (s) |
| 50 | 0 | 0.96 | 1.85 | 75 | 0 | 1.21 | 1.62 |
| 50 | 50 | 1.24 | 2.02 | 75 | 50 | 1.31 | 1.86 |
| 50 | 100 | 1.19 | 1.73 | 75 | 100 | 1.11 | 2.06 |
| 50 | 200 | 1.31 | 1.36 | 75 | 200 | 1.32 | 2.14 |
| 50 | 300 | 1.25 | 1.51 | 75 | 300 | 1.41 | 1.93 |
| 50 | 400 | 1.11 | 1.42 | 75 | 400 | 1.30 | 1.37 |

Figure 9 depicts the responses of the participants comparing the overall stability of the robot–human handovers using the different sets of the impedance parameters. It can be observed that 11 participants (55 %) and three participants (15%), respectively, were significantly satisfied and very satisfied with the overall stability of the robot handover system using the first parameter set. The second set shows 12 (60%) and 6 (30%) subjects were satisfied and very satisfied. There is only one person (5%) very dissatisfied with the parameter set 1, while another leads participants to experience the robot handover without any dissatisfaction. To compare the mean measurements of the two set undertaken from the same sample group, the paired samples t-test was statistically utilized. The following hypotheses $H_0$ and $H_1$ were tested as:

$H_0 : \mu_{Set2} = \mu_{Set1}$, which means the median difference between pairs of observations is zero so that there is no significant difference in the population distributions.

$H_0 : \mu_{Set2} > \mu_{Set1}$, which means the median difference between pairs of observations is not equal to zero and the population rating from set 2 is greater than that of set 1.

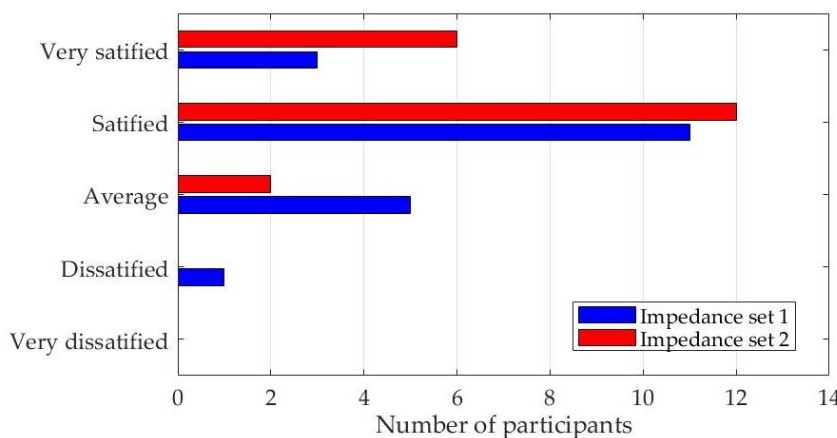

**Figure 9.** Survey responses for the human–robot handover tasks.

The results of the statistical analysis based on the paired $t$-test present that ratings of the robot performance based on the parameter set 1 and 2 were 3.80 and 4.20, respectively, with corresponding standard deviations of 0.768 and 0.616, as summarized in Table 3. This represents the test subjects were more appreciative of the performance of the robot behavioural control using the second parameter set rather than the first set. The value of the paired-sample test was $-3.559$ (2-tails significance = 0.002). It can be described that the hypothesis $H_0$ has to be statistically rejected and the alternative hypothesis $H_1$ accepted. This means that there is a significant difference in the population distributions from the participants. Additionally, the population rating from set 2 is greater than that of set 1, in which the participants were more comfortable with the robot's stability using the

impedance parameters made up of $K_d$ (50), $B_d$(75) and $M_d$(0.25), more than those of the first set.

**Table 3.** T-test results of the comparison between the parameter set 1 and 2.

| Paired Samples Statistics | | | | | |
|---|---|---|---|---|---|
| | | Mean | N | SD | SD Error Mean |
| Pair 1 | Impedence1 | 3.80 | 20 | 0.768 | 0.172 |
| | Impedence2 | 4.20 | 20 | 0.616 | 0.138 |
| **Paired Samples Correlations** | | | | | |
| | | | N | Correlation | Sig. |
| Pair 1 | Impedence1 & Impedence2 | | 20 | 0.757 | 0.000 |

| Paired Samples Test | | | | | | | | |
|---|---|---|---|---|---|---|---|---|
| | | Paired Differences | | | | | | |
| | | | | | 95% Confidence of the Difference | | t | df | Sig. (2-tailed) |
| | | Mean | SD | SD Error Mean | Lower | Upper | | | |
| Pair 1 | Impedence1 & Impedence2 | −0.400 | 0.503 | 0.112 | −0.635 | −0.165 | −3.559 | 19 | 0.002 |

Finally, the results of the robotic dynamic response based on the hybrid PID position and impedance control in the robot-to-human handovers were carried out and then compared with the outcomes of the HHH. Figure 10 shows an example of a comparison between the interactive force profiles against time from the HHH and HRH tests. It can be concluded that the quantitative measurement of the performance of robot behavioural control can be considered acceptable for object handover interaction with a human receiver. It provided effective performance during the natural cooperative tasks, where the robot was able to successfully pass the object to the human in a safe, reliable and timely manner. However, after careful analysis with regard to the HRH results, the robot has to perform a slightly longer handover period than that of the HHI, even though the maximum forces from both cases are similar. This is because the robot control algorithm was initially designed to ensure its safe and natural operational characteristics specifically while physically releasing the object to the receiver. Then, this might result longer object handover process.

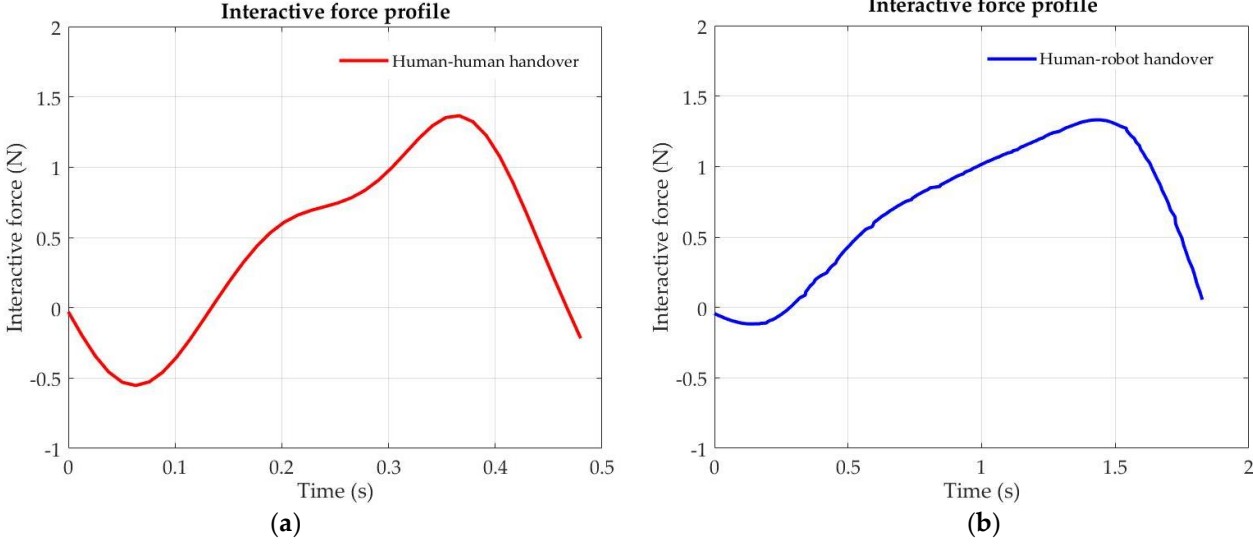

**Figure 10.** Interactive force profiles against the time of (**a**) the HHH test and (**b**) the HRH test.

Consequently, it can be concluded that understanding kinematically and dynamically how two humans physically collaborate while naturally performing object handover tasks offers the effective performance and reliability of the robot-to-human handover procedure and acceptable HRI implementation. This was in agreement with the parallel test outcomes, demonstrating significant satisfaction with the overall performance of the HRH system, as measured by an average rating of 4.20 on a five-point scale.

## 4. Conclusions

This paper contributes to the implementation of a human-like behavioural control strategy for seamless HRH. The robotic conceptual framework was established by understanding the principle of human haptic interaction when two humans work together in a joint effort to complete an object handover task. The performance of the robot–human handover system has been evaluated based on the interactive fore profiles, handover time and how comfortable the participants felt in participating in the HRH tasks. The results provide effective performance during the HRH, where the HSR robot was able to successfully pass the object to the human effectively. The safety systems proposed are working successfully and thus avoiding the likelihood of unsafe HRH actions being taken. Moreover, the feedback responses from the human subjects participating in the HRI also agreed with the parallel test results that gave more satisfaction with a higher average rating scale.

Therefore, it can be concluded that the robot based on the developed behavioural control algorithm is capable of tracking the object transfer location, manipulating the object and handling it to the receiver in a safe, reliable and dexterous manner. This developed robotic system can enhance the accuracy of positioning and force regulating abilities in the HRH system and also increase the smoothness of physical HRI. The outcomes of this research can successfully play a beneficial role in the daily lives of people in future.

**Author Contributions:** Conceptualization, P.N.; methodology, P.N. and T.S.; software, T.S.; validation, P.N. and T.S.; formal analysis, P.N. and T.S.; investigation, P.N. and T.S.; resources, P.N.; data curation, P.N. and T.S.; writing—original draft preparation, P.N.; writing—review and editing, P.N.; visualization, P.N.; supervision, P.N.; project administration, P.N.; funding acquisition, P.N. All authors have read and agreed to the published version of the manuscript.

**Funding:** This research received no external funding.

**Institutional Review Board Statement:** The study was conducted according to the guidelines of the Declaration of Helsinki.

**Informed Consent Statement:** Informed consent was obtained from all subjects involved in the study.

**Data Availability Statement:** Not applicable.

**Conflicts of Interest:** The authors declare no conflict of interest.

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
