# Peer review of "A Human-Inspired Control Strategy for Improving Seamless Robot-To-Human Handovers"

_applsci, doi:10.3390/app11104437_

Round 1
Reviewer 1 Report
This article deals with the development of an appropriate set of behaviour-based control for robot-to-human object handover by first understanding an equivalent human-human handover. The topic is novel and interesting for social and industrial collaborative robotics. The kind and number of references is adequate. The texts are comprehensible even if a general language check for correcting errors and missing punctuation is suggested. In general, the explanation of the work is clear even if there are some changes to be considered for the improvement of the article, especially from the point of view of the structure. In general, all the elements needed for the description of the research are present but divided in a not optimal way. Therefore, it is suggested to re-.structure the work by clearly defining: an introduction, a review of the literature, a section related to materials and methods (also describing the experimental set-up and development), a section related to the explanation of the results, and finally a discussion and conclusions section. Following, some other minor suggestions are provided:
Abstract
Line 18: some results about a survey are described without first introducing the fact that a survey has been developed for a certain purpose. It is suggested to improve this part by also introducing the scope of the survey with respect to the overall experiment before explaining its results.
- Introduction
- Line 28: please consider changing “most challenges” with “main challenges”;
- Line 33-35: this sentence seems to be unclear to the reader. Please rephrase;
- Line 36: “Human-human handovers” has been previously defined as HHH, so please use the acronym for its definition;
- Line 48: please consider changing with “Neranon et al.”;
- Line 60: please consider changing with “Chien-Ming et al.”;
- Line 68: it misses the point after “etc.)”
- Line 73: please consider changing with “Yi et al.”;
- Line 74: please define “HSR”;
- Line 97: what does it mean that a HRS “was used in this project”? What is the role of such a robotic system in the proposed research? If the purpose of its use was to gain more knowledge about the system for further activities, this concept has to be summarized in a short sentences and better clarified;
- Background of HHH Strategy
- Line 121: please consider changing “transfer” with “transferring”;
- Line 127-153: this section is interesting and well explained. Nevertheless, these concepts should be summarized as they are key findings of previous works and only serve as a theoretical base for the development of the HRH strategy. Please also consider just mention them by using a couple of sentences and related references;
- The Robotic human-inspired control strategy for HRH
- Line 154-162: this part seems to be unnecessary;
- Line 202: is it correct “HRI”?;
- Line 208: is it correct “Section 3.2”?;
- Line 219: the sentence is not clear. Please rephrase;
- Line 240: please remove a couple of brackets;
- Line 296: “flowing” or “following”? Please check;
- Line 303: the acronym “HRI” has been defined in the introduction. Please use it instead of the full words. Please also check the whole article since it seems that many times the use of acronyms and related full-words are misused;
- Line 353: please add a space between “mass” and “m0”;
- Line 357: please add a space between “and” and “mk”;
- Equation 23: it is suggested to present the matrix before Fig.7 to be continuous with the information provided in the text;
- Section 3.7: please move the title to the same page of the text;
- Line 397: it could be useful to add more details about the definition of the safety limits. How were these limits defined? Do you also consider the indications provided by recognized documentation, e.g. by ISO TS 15066 for mechanical risks in human-robot collaboration? Even if it is related to the industrial sector, you can find some useful information in this technical specification also for social robotics.
- Evaluation of the robot-human handover system
- Line 454: please add a space between “as” and “kp”;
- Section 4.2: please move the title to the same page of the text;
- Figure 8, Figure 10 and Figure 11: the font is different with respect to the one used in the texts. Please check;
- Figure 9: it is suggested to cancel the lower part which represents the video player;
- Line 486-488: please check the meaning of the sentence and rephrase;
- Line 493: please add a space after “400”,
- Table 1: please move the heading and the first row to the same page as the rest of the table;
- Line 503: it is suggested to add a couple of sentences describing the structure of the survey (e.g. type and number of questions used, etc.);
- Line 505: please check the meaning of the sentence and rephrase;
- Line 541: please add a space after “kd”;
- Table 3: it is not mentioned in any text of the article. Please check and add its description in the text;
- Conclusions
- Line 578: please add a space after “Bd”;
- This section could be improved by describing more in detail the main contribution of the findings of this work with respect to the research in the field of social robotics and human-robot interaction;
General remarks
- Sometimes capital letters have been used for some words of title texts, sometimes not. Please check;
- In general, the article is comprehensible. Nevertheless, it is suggested a general language check for correcting errors and missing punctuation;
Author Response
To Review 1
First, we would like to sincerely thank you for your useful recommendations to improve my manuscript. All the comments we received on this study have been taken into account in enhancing the quality of the article, and we present our reply to each of them separately. We have attached the revised paper as attached.
Best regards,
Authors

Reviewer 2 Report
Dear authors, your article undoubtedly deserves publication and will arouse the interest of a certain circle of interested persons, however, I have a number of comments to you, namely:
1. The list of references should be significantly expanded, since the issues of human-robot interaction are extensive and multifaceted.
2. Figure 2 (b) and Figure 6 the title does not reflect the essence. Rather, they are functional diagrams.
3. Why the issue of using a PID controller with fuzzy control has not been considered.
Author Response
To Reviewer 2
First, we would like to sincerely thank you for your useful recommendations to improve my manuscript. All the comments we received on this study have been taken into account in enhancing the quality of the article, and we present our reply to each of them separately. We have attached the revised paper as attached.
Best regards,
Authors
